# Does Coastal Local Government Competition Increase Coastal Water Pollution? Evidence from China

**DOI:** 10.3390/ijerph17186862

**Published:** 2020-09-20

**Authors:** Weiteng Shen, Qiuguang Hu, Xuan Yu, Bernadette Tadala Imwa

**Affiliations:** 1School of Business, Ningbo University, Ningbo 315211, China; 1801092003@nbu.edu.cn (W.S.); 1511091887@nbu.edu.cn (X.Y.); bernaimwa@yahoo.co.uk (B.T.I.); 2East China Sea Institute, Ningbo University, Ningbo 315211, China

**Keywords:** local government competition, coastal water pollution, financial pressure, fixed-effects panel regression models

## Abstract

China has formulated many policies and regulations for the management of the coastal water environment. However, the coastal water environment has not been significantly improved. The perspective of local government competition can provide an explanation for this phenomenon. This study uses panel data comprising 48 coastal cities in China from 2004 to 2017 as bases explore the impact of coastal local government competition on coastal water pollution by using a two-way fixed-effects panel regression model. Results show that coastal local government competition increased coastal water pollution. However, a sub-sample estimation based on fiscal pressure shows that coastal local government competition only increased the coastal water pollution of the high fiscal pressure group, and its impact on the coastal water pollution in the low financial pressure group failed to pass the significance test. In addition, sub-sample estimation based on different periods shows that the impact of coastal local government competition on coastal water pollution was not significant before 2008, but showed a significant positive impact after 2008.

## 1. Introduction

The ocean is the blue home on which mankind depends, particularly with its rich resources and strong pollution absorption capacity. However, the continuous increase of human and economic activities has extensively damaged the marine environment [1], particularly the coastal waters directly touched by human activities. In 2002, the water quality of category IV seawater and inferior category IV seawater accounted for 35.9% in China’s coastal waters. Such seawater cannot be directly touched by humans, and cannot meet the basic conditions for the survival of the majority of organisms. This proportion generally declined after over 10 years of governance. However, by 2018, the proportion continued, maintaining a high level (18.7%). China’s coastal areas have varying degrees of pollution in coastal waters and bays. In 2017, among the 11 provinces or municipalities along China’s coast, the quality of the coastal waters in five provinces or municipalities was extremely poor or poor, and the water quality of estuaries in six provinces or municipalities was moderate or severely polluted [2]. Of the 44 bays above 100 square kilometers, 20 bays are below grade IV water quality in all seasons [2]. The current marine environmental pollution has become one of the main obstacles for the high-quality development of China’s economy. To reverse the deteriorating trend of the marine environment, in 2018, the State Council established the Ministry of Ecological Environment to integrate the marine environmental protection responsibilities that were originally dispersed in the Ministry of Environmental Protection, State Oceanic Administration, and other similar agencies. Although Institutional reforms have systematically resolved the “water control in Kowloon” issue in coastal water environmental governance, many issues within the organization should be clarified. Moreover, the problem of incentive distortion behind coastal water pollution still needs to be further resolved.

For the coastal water environment, China has adopted the principle of territorial governance. That is, coastal local governments are responsible for the environmental governance of coastal areas under their jurisdiction. Therefore, local government actions may be a clue to the environmental problems in China’s coastal waters. Local governments have multiple goals, such as social stability, economic development, and environmental protection. These goals constitute the objective function of local government officials. Promotion championship theory states that under various objectives, local officials will only focus on indicators that can be evaluated, and indicators that are not in the evaluation range or difficult to measure will be disregarded [3]. Therefore, Gross Domestic Product (GDP) has become a key indicator sought by local governments. This self-selection of goals makes many actions of the local government revolve around economic growth. Local officials maximize local economic growth to obtain scarce promotion opportunities and evolve into fierce competition for economic growth among regions. Under the economic incentive, local governments disregard the protection and governance of the coastal environment and even have the motivation to destroy the coastal environment. According to the feedback to provinces and municipalities from the National Marine Inspection Team’s in July 2018, the environmental protection department of Zhejiang Province reported only 462 sewage outfalls to the sea. However, after actual surveys, the province has 1376 sewage outfalls to the sea. Shanghai has provided 98 land-based pollution sources that entered the sea, and actual inspection revealed that there were 148 land-based pollution sources. Moreover, coastal provinces and municipalities such as Shanghai, Tianjin, Zhejiang, and other provinces generally have illegal reclamation and illegal use of the sea. Given the preceding facts, local economic competition is an important perspective to understand coastal water pollution and its governance issues.

The current study contributed to this field of study as follows. First, by incorporating local government actions into the analysis of pollution causes in China’s coastal waters, this study analyzes the spillover effects of economic competition among local governments on coastal water pollution. On this basis, the heterogeneous impact of local government competition on China’s coastal water pollution was tested based on grouping financial pressures and time periods. Second, previous studies have mainly used province-level data to study China’s coastal water pollution problems. By contrast, the current research uses prefecture-level data to analyze coastal water pollution, thereby considerably reflecting the importance of local competition in China’s coastal water pollution control. Lastly, this study uses the pollutant concentration collected manually from the China Coastal Environmental Quality Bulletin (CCEQB) to measure the degree of coastal water pollution instead of pollutant emission data used in many other studies.

This remainder of this paper is organized as follows. Section 2 provides literature reviews and proposes of the related hypotheses. Section 3 introduces the methodology and data used to test the hypotheses. Section 4 provides the empirical results. Section 5 presents the heterogeneity analysis. Section 6 provides the discussion. This paper also concludes with a conclusion and policy implications.

## 2. Literature Review and Hypotheses 

### 2.1. Literature Review

#### 2.1.1. Research on Local Government Competition 

Local officials have the “politician” attribute and inherent incentives to pursue political promotion, thereby making local governments show competitiveness in taxation and fiscal expenditure [4]. The literature on local government competition has focused on the economy and environment. In terms of the economy, research from China has shown that competition can incentivize local governments to pursue economic construction and promote local economic development [5,6]. On the one hand, local governments attract numerous companies to invest locally through tax cuts and low prices of industrial land [7,8]. On the other hand, a large amount of fiscal expenditure is allocated to infrastructure construction [9,10]. Competition among local governments also has spillover effects on the environment. Under the principle of “GDP first”, competition has evolved into economic competition, and extensive resources have been invested in infrastructure construction and industrial development, thereby disregarding investments in environmental public goods [11]. Jun et al. [12] used the difference between the national and local effective tax rates to measure tax competition and found that local governments through various preferential policies to attract investment have aggravated the degree of environmental pollution. Bai et al. [13] found that inter-provincial tax competition increases local environmental pollution, and worsens the environment in adjacent areas. However, some studies have presented different stories. Jie and Wen [14] showed that local government competition will increase industrial wastewater discharge, but reduce industrial waste gas and solid waste discharge. Tian et al. [15] used Chinese provincial data to explore the impact of tax competition on the environment and showed that with an increase in tax competition, industrial pollution emissions will gradually decrease. At present, no research has involved the impact of local government competition on the coastal water environment.

#### 2.1.2. Research on the Causes and Governance of the Coastal Water Environment

As people have gradually understood the importance of the ocean, considerable focus has been given to the marine environment and many studies involving the sources and causes of coastal water pollution have emerged. Coastal water pollution comes from land-based pollution, marine development and engineering construction, petroleum exploration, and marine accidents, among which land-based pollution is the main source [16,17]. Land-based pollution is mainly caused by the continuous expansion of human economic activities. In the research involving China’s coastal water pollution, the main indicators of coastal water pollution are whether the water quality meets the corresponding standards, the amount of industrial wastewater discharged directly into the sea, and the volume of industrial solid wastes discharged directly into the sea. On this basis, these studies examine the relationship between economic development and coastal water pollution. Chen et al. [18] used the above indicators to measure pollution in China’s coastal waters and examined the relationship between human economic activities and coastal water pollution, and showed that coastal water pollution and economic growth show an “inverted N-shaped” relationship. Xu and Wang [19] used the entropy variable equation to test the degree of coordination between economic activities and the coastal water environment and determined that no coordination relationship exists between the two aspects in the majority of the cases.

With the increasing risk of marine pollution, numerous studies have focused on the control of marine pollution. Marine environmental pollution may span multiple countries or regions, and the cooperation of multiple countries or regions is required to achieve effective governance [20]. At present, the research on marine environmental governance focuses on how to solve transboundary issues based on a global perspective [21,22]. Ding [23] studied the transboundary problems of marine pollution within the Association of Southeast Asian Nations (ASEAN) and found that the top-down governance model cannot solve the marine pollution problems faced by ASEAN. Relative to marine pollution in the high seas, coastal water pollution is mainly within the scope of a country or region, spanning multiple administrative regions within a specific geographic area. The existing literature on coastal water pollution has mainly studied the control of coastal water pollution from the three perspectives of market, government, and society [24]. Cadman et al. [25] studied the role of NGOs as a social force in the governance of Canadian sea areas. They believed that environment NGOs (ENGOs) have bridged the interaction between the government and stakeholders and passed the scientific data and information generated by the researchers to decision-makers. From the perspective of government governance, Yu [26] explained that China’s coastal water environmental governance should rely on the joint action of the central and local governments to achieve effective governance. Hu et al. [24] used the analysis of the causes of China’s coastal water environment as a basis to explain that, in addition to social and government forces, the effective management of coastal water pollution also requires market forces.

Through a review of the literature, we can find that the existing research on local government competition and coastal water pollution is relatively extensive. These studies have analyzed local government competition and coastal water pollution from different perspectives and obtained meaningful conclusions. At present, the existing research on pollution in China’s coastal waters is mainly based on the assumption that the government is selfless. However, research on public choice and modern fiscal federalism theory shows that individuals in political markets and political decision-making have the characteristics of economic men, and they pursue the maximization of their own interests rather than that of social welfare [27,28]. In an institutional environment that competes for growth, local governments have a strong incentive to sacrifice the coastal water environment in exchange for economic growth. The government competition perspective can provide a new explanation for the deteriorating coastal water environment of China. On the basis of the assumption that local government officials are economic men, this study uses data from China’s coastal cities to study the impact of local government competition on coastal water pollution.

### 2.2. Research Hypotheses

Different countries have adopted different governance systems for environmental pollution. Eastern and southeast Asian countries generally adopt a “paternalistic” governance pattern. Given the complexity of environmental issues, this governance model may be more efficient than the “democratic” governance model [29]. In China, the control of coastal water pollution mainly depends on local governments. However, studies of public choice and modern fiscal federalism theories have shown that similar to individuals engaged in economic activities, local officials also seek to maximize their own interests rather than that of social welfare [28]. Therefore, local governments will not necessarily play a role in promoting governance of the coastal water environment. Economic growth is the core evaluation index for the promotion of local officials in China, and the chief officials of coastal cities are often promoted through economic competition with other cities in the same province. On the one hand, to achieve short-term rapid economic growth, local governments will relax local environmental standards to attract investments from enterprises, and allow enterprises to permit pollution emissions in production [30], thereby forming a so-called “government-enterprise collusion” [31]. This type of “race to the bottom” on the environmental regulation has caused numerous untreated pollutants to be directly discharged into the sea. On the other hand, under the pressure of local competition, coastal local governments should obtain funds through land transfers to support economic development. However, the available land area in China’s coastal cities is decreasing with the advancement of urbanization. To expand the decreasing land, local governments have approved numerous reclamation projects without environmental impact assessments, thereby damaging the coastal water environment and reducing the self-purification capacity of the coastal ecosystem. Accordingly, Hypothesis 1 is proposed on the basis of the preceding analysis.

**Hypothesis** **1.**
*Coastal local government competition will increase coastal water pollution.*


For coastal cities facing different fiscal pressures, the impact of local government competition on coastal water pollution may vary. On the one hand, fiscal pressure will affect resource allocation and crowd out environmental public goods investment. By considering the scarcity of financial resources, officials in cities experiencing immense fiscal pressure will use fiscal funds to best reflect the performance of the government to obtain high marginal returns. Meanwhile, the control of coastal water pollution has a strong externality, and the benefits and costs are not equal. Therefore, cities with immense financial pressure will allocate as many financial funds as possible to economic development and reduce investments in coastal water environmental governance to maximize their political achievements. On the other hand, cities facing financial pressure will considerably focus on the output value of enterprises and their contribution to finance while disregarding the environmental impact. The scarcity of financial resources will prompt local governments to introduce enterprises that can substantially contribute high GDP to the local government. Therefore, industrial enterprises that can bring high GDP and generate considerable tax revenue will become the focus of local government introduction. However, industrial enterprises often cause high water pollution. The pollution generated by industrial enterprises is often directly discharged into the coastal waters, thereby harming the coastal water environment. Thus, Hypothesis 2 is proposed as follows.

**Hypothesis** **2.**
*The impact of coastal local government competition on coastal water pollution will vary in cities with high fiscal pressure and cities with low fiscal pressure.*


## 3. Methodology and Data

### 3.1. Model Settings

This study focuses on the impact of coastal local government competition on coastal water pollution. We use the fixed-effects panel regression model to test hypotheses 1 and 2. The constructed model is as follows:(1)$$lnorganicN_{it}=\beta_{0}+\beta_{1}lnperfdi_{it}+\overset{8}{\underset{j=2}{\sum }}\beta_{j}Control_{it}+\gamma_{p}+\delta_{t}+\varepsilon_{it}$$
where $lnorganicN_{it}$ is the inorganic nitrogen concentration of coastal city *i* in year *t*, which is used to measure coastal water pollution. $lnperfdi_{it}$ is the natural logarithm of foreign direct investment (FDI) per capita of the coastal city *i* in year *t*, which is used to measure the degree of local competition; $Control_{it}$ is the control variable; and $\gamma_{p}$ is the province fixed effect, which is used to control the province characteristics that do not change with time. The reason for controlling the provincial effect is that the main competitors of coastal cities are other cities in the same province, and the essence is the competition between government officials. Under a limited promotion quota, the chief officials of coastal cities will exert their best effort to cater to the institutional arrangements of the provincial government to obtain promotion capital. Therefore, the competition shown by coastal cities will have a strong correlation with the institutional arrangements of the provincial government. If we do not control the province effect, then it will cause a missing variable bias. $\delta_{t}$ is the time-fixed effect, which is used to control the effect of special events that occur in different years; and $\varepsilon_{it}$ is an i.i.d disturbance term. $\beta_{0}$ is the estimated value of the intercept term. $\beta_{1}$ to $\beta_{8}$ are the estimated coefficients of variables.

### 3.2. Variable Description

#### 3.2.1. Dependent Variable

The dependent variable in this study is coastal water pollution, which is measured by inorganic nitrogen concentration. A few studies involving China’s coastal water pollution have used the discharge amount of wastewater, percentage of points in excess of the pollution standard, or discharge amount of industrial wastewater per unit of marine economic output to measure coastal water pollution [32,33]. However, these indicators of pollution emissions do not actually reflect the degree of coastal water pollution. The percentage of points in excess of the pollution standard is a proportion concept and cannot reflect the average situation of coastal pollution in coastal cities. The degree of coastal water pollution should be expressed as concentration. CCEQB published data on the concentration of various pollutants. By considering the availability of data, this study selected the inorganic nitrogen concentration to measure the coastal water pollution status. That is, the higher the concentration, the more serious the coastal water pollution.

The reasons for using inorganic nitrogen concentration to measure coastal water pollution are as follows. First, CCEQB indicated that the major pollutants in China’s sea areas are inorganic nitrogen. Second, the distribution of inorganic nitrogen concentration in coastal provinces approximates that of the coastal pollution in China. Figure 1 shows the distribution of inorganic nitrogen concentration in 11 coastal provinces. Evidently, Shanghai and Zhejiang have the highest concentrations, while Guangxi and Hainan have the lowest concentrations. Moreover, CCEQB included the water quality assessment results of 11 provinces along the coast of China. The evaluation results over the years have shown that Shanghai and Zhejiang had the worst water quality, while Guangxi and Hainan had the best water quality [34]. Therefore, the distribution of inorganic nitrogen concentration is consistent with the provincial-level water quality. We used the preceding two reasons as bases to state that inorganic nitrogen concentration can reflect coastal water pollution in China.

#### 3.2.2. Core Explanatory Variable

The core explanatory variable in this study is local government competition. Since the reform and opening up in 1978, the Chinese government has constantly regarded attracting foreign investment as one of the core government tasks to improve the level of investment and promote economic development. Under the assessment system of GDP-oriented local officials, attracting foreign capital has become the main means for local governments to compete. Therefore, this study uses the logarithmic value of per capita actual utilization of FDI (hereinafter referred to as “actual FDI per capita”) to measure local government competition based on China’s actual background and existing research [9,29,35]. The higher the actual FDI per capita, the stronger the competition among local governments. To ensure the robustness of the estimation results, the current study uses two types of indicators to measure the degree of local government competition in the robustness test.

#### 3.2.3. Control Variables

To control for the influence of other factors on the estimated coefficients of the core explanatory variables to substantially reduce the missing variable bias, we drew on the studies of Bai et al. [13] and Ding et al. [36] and considered the characteristics of China’s coastal water pollution. Moreover, we select the following control variables. (1) Level of economic development (LNPERGDP)—the impact of economic development level on environmental pollution has been confirmed by numerous studies [37,38,39]. We used the logarithm of GDP per capita to measure the level of economic development and added the quadratic term of the logarithm of GDP per capita to test whether the environmental Kuznets curve exists. (2) Population density (LNDENSITY)—the increase in the population density of coastal cities has caused an increase in pollution emissions, and numerous pollutants have flowed into coastal waters through the surface and underground runoff, thereby leading to increased coastal water pollution. We used the logarithm of population density as a proxy variable for population density. (3) Technological innovation (LNPATGRANTED)—technological innovation can improve the existing production technology, thereby reducing the pollutants discharged in production and benefiting the improvement of the coastal water environment. We used the logarithm of the total number of patents to measure technological innovation. (4) Urbanization—on the one hand, the increase in urbanization level has made the population concentrated in cities and towns, thereby resulting in scale effects and increased pollution emissions. On the other hand, urbanization will result in agglomeration effects, improve the efficiency of pollution control, and reduce the impact of pollution emissions on the coastal water environment. Therefore, the impact of urbanization on coastal water pollution depends on which effect of scale and agglomeration is stronger. We used the proportion of the non-agricultural population to the total population to measure urbanization. (5) Industrial structure (SECONDGDP)—industries of coastal cities are mainly located in coastal areas for sewage discharge. Numerous types of untreated industrial pollution are directly discharged into the sea, causing coastal water pollution. This study uses the proportion of secondary industry to GDP as the proxy variable of the industrial structure to control the impact of industrial structure on coastal water pollution. (6) Marine economic activity (mgdp)—although coastal water pollution is mainly manifested as land-based pollution, marine economic activities, such as marine resource exploitation, mariculture, and marine transportation, also affect coastal water pollution. We use the logarithm of gross ocean product as a proxy variable for marine economic activity.

### 3.3. Data Sources

Inorganic nitrogen concentration for coastal cities collected from the CCEQB. Prior to 2017, CCEQB was issued annually by the former Ministry of Environmental Protection of China. After the adjustment of government agencies in 2018, the CCEQB in 2017 was issued by the Ministry of Ecology and Environment of China. The CCEQB mainly includes annual indicators reflecting the ecological environment of China’s coastal waters. The concentration of inorganic nitrogen in the coastal waters in CCEQB is the monitoring data of the nationally controlled environmental quality points. The monitoring time is divided into three periods according to the water period. The first period is from April to May (a high-water period), the second period is from July to August (common water period), and the third period is from October to November (dry period). The inorganic nitrogen concentration data in this study is the average of the inorganic nitrogen concentration in three periods. We chose the inorganic nitrogen concentration in coastal waters from 2004 to 2017 for the following reasons. Firstly, CCEQB and China Marine Environmental Status Bulletin were merged into the China Marine Ecological Environmental Status Bulletin after 2017. Pollutant concentration is no longer reported in the combined bulletin. Secondly, the serious lack of data before 2004. In addition, considering the substantial lack of data in some coastal cities, 48 coastal cities were eventually selected as the research sample, accounting for approximately 89% of the total coastal cities.

Foreign direct investment (FDI), the proportion of secondary industry in GDP, GDP per capita, and population density are collected from the China City Statistical Yearbook. The total population and non-agricultural population data used to calculate urbanization, local general public budget revenue, and local general public budget expenditure were from the China City Statistical Yearbook and National Economic and Social Development Statistical Bulletins. The number of patent grants at the end of the year came from the China Research Data Service Platform. The gross marine product of coastal cities was lacking and replaced by the gross marine product of coastal provinces, which were sourced from the China Marine Statistical Yearbook. In the robust test, the proportion of cross-sections worse than grade V water quality in all watersheds were from the China Environmental Statistics Yearbook, and the chemical oxygen was sourced from the CCEQB. All currency variables are adjusted to 2004 constant prices using the corresponding price indices. Descriptive statistics of all variables are presented in Table 1.

Table 1 shows that the maximum value of inorganic nitrogen is 2.271 mg/L, which is 8 times the average value. The city with an inorganic nitrogen concentration of 2.271 is Jiaxing City in Zhejiang Province. First, the sea area where this city is located belongs to the East China Sea, which is the most polluted sea area among all sea areas in China. Second, Jiaxing City is located in Hangzhou Bay, the most polluted bay in the East China Sea. These two facts make the inorganic nitrogen concentration in Jiaxing extremely high.

## 4. Empirical Results

### 4.1. Benchmark Estimation Results

This study used the fixed-effects panel regression model to test the impact of local government competition on coastal water pollution. Table 2 shows the estimation results for the two-way fixed-effects panel model. Moreover, the regression results without adding fixed effects are given to compare it with the estimation results of the fixed-effects panel model.

Columns (1) and (2) present the regression results without taking the fixed effects of provinces and years. The estimation results show that without the addition of control variables, the estimated coefficient of LNPERFDI is significantly positive at the 5% level, thereby indicating that local government competition will increase coastal water pollution. After adding the control variables, the estimated coefficient of LNPERFDI decreased but remained significantly positive at the 5% level. Columns (3) and (4) show the regression results that include the fixed effects. After including the fixed effects, the model’s goodness of fit was substantially improved, and the sign and significance of the LNPERFDI’s estimation coefficient did not change. The preceding estimates indicate that increased local government competition intensity will increase coastal water pollution. Therefore, Hypothesis 1 is verified. In the competition of local governments, coastal cities rely on the location advantage of being close to the sea to develop heavy and chemical industries. Many coastal cities have launched large-scale heavy industry projects to exploit marine resources. In Zhanjiang, Guangdong, the government approved two large projects: Baosteel Steel Base Project and a new oil refining integrated branch project, Tianjin is building a new chemical park and port industrial zone [36].

We further explain the estimation results of the control variables in Column (4). The estimated coefficients of LNPERGDP and its quadratic term are negative and positive, respectively, and both pass the statistical significance test at the 1% significance level. The result shows a U-shaped relationship between ocean economic development and coastal water pollution. The coefficient of LNDENSITY is significantly positive at the 1% level. That is, as the population density of coastal cities continues to increase, coastal water pollution will continue to increase. The coefficient of LNPATGRANTED is positive but did not reach statistical significance. The estimated coefficient of URBANIZATION is significantly positive. This result shows that the scale effect brought by urbanization plays a key role. With continuous urbanization, coastal water pollution continues to increase. The coefficient of SECONDGDP is significantly positive, which is consistent with expectations. The secondary industry will bring extensive pollution. In coastal cities, these pollutants are often directly discharged into the sea, resulting in increased coastal water pollution. Therefore, the higher the proportion of the secondary industry in GDP, the more serious the coastal water pollution. The impact of marine economic activities on coastal water pollution did not reach statistical significance.

### 4.2. Robustness Checks

To ensure the reliability of the estimation results, this study further conducts a robustness test by replacing proxy variables, removing interference samples, removing outliers, and considering missing variables.

#### 4.2.1. Replace Proxy Variables

In this subsection, we conduct a robustness test by replacing the proxy variables of the explained and core explanatory variables. First, we use chemical oxygen demand (COD) as a proxy variable for coastal water pollution. High COD means that the water contains numerous reducing substances, mainly organic pollutants. The higher the COD, the greater the pollution of organic matter in the water. Before 2015, CCEQB included the average concentration of chemical oxygen demand (COD) in the coastal areas of coastal cities. Therefore, we re-estimated Model (1) by using the COD concentration and samples before 2015. The estimated results are shown in Column (1) of Table 3. The estimation results show that the direction and significance of LNPERFDI’s estimated coefficients did not change significantly.

Second, we use two types of indicators to measure the local government competition. The setting of the first indicator refers to Deng et al. [30], and the proportion of the actual utilization of FDI in cities to the actual utilization of FDI in the province to which the city belongs (PROFDI) is used to measure local government competition. In addition, we use the promotion incentives of coastal city officials to measure local government competition. Zhou [3] explained that to achieve promotion, local government officials compete with GDP as the core, which is manifested as the competition among local governments. That is, local government competition is partially driven by the promotion incentives of local officials. Under China’s official appointment system, the promotion of officials faces age restrictions. When an official exceeds a certain age, the possibility of a promotion will be significantly reduced. Therefore, this scenario provides a good opportunity for us to find proxy variables for local government competition. We take 55 as the cut-off point by referred to Yu et al. [40] and Wu and Zhou [41]. Officials above and below 55 years are defined as low and high-promotion incentive officials, respectively. Local governments in cities where high-incentive officials are located are highly competitive, whereas governments in cities with low-incentive officials have weak competition. Column (3) in Table 3 provides the estimated results obtained by the municipal party secretary promotion incentives (DUSAGE) to measure local competition. Column (4) presents the estimated results obtained by the mayor promotion incentives (DUMAGE) to measure the local government competition. The estimation results of Columns (3) and (4) show that the estimated coefficients of DUSAGE and DUMAGE are positive, and both can pass the statistical significance test. The preceding results show that Hypothesis 1 is supported in the case of replacing the proxy variables with the explained and core explanatory variables.

#### 4.2.2. Other Robustness Checks

In addition to replacing the proxy variables of the dependent and core explanatory variables, we also implemented three types of robustness tests. First, the interference samples were removed. Among the coastal cities, Tianjin and Shanghai are municipalities directly responsible for the central government and have higher administrative levels than other cities. Accordingly, the constraints they face are significantly different from those in other cities. Given that the sample includes Tianjin and Shanghai, which may have an impact on the estimation results, the samples of these two cities were eliminated. Second, variables were winsorized. The inorganic nitrogen concentration of coastal waters used in this study belongs to monitoring data, and abnormality may occur during monitoring. Thus, the samples were arranged from small to large, and winsorization was performed in the first 1% sample and the last 99% sample. Third, a discussion on the missing variable bias was conducted. Coastal water pollution in China mainly comes from land-based pollution. In addition to pollution caused by coastal cities, a large part of pollutants flows into the ocean through the transport of various river basins. Pollution from the river basin is a key factor affecting coastal water pollution. Moreover, a large extensive pollution from river basins may also affect the competition among coastal cities. The governance of coastal water pollution often depends on watershed pollution control (i.e., the need for overall watershed management). In the absence of integrated watershed management, the marginal net benefits of governance investment from coastal cities will be diluted by watershed pollution. Hence, coastal cities will not adopt effective governance and protection measures. The behavior of coastal cities may have some correlation with the pollution level of river basins. The correlations between river basin pollution and coastal water pollution and between river basin pollution and local government competition can leads to missing variable bias. The proportion of worse than grade V water quality was used as a proxy variable for watershed pollution (BASINPOLL) to solve the missing variable bias. The results of the above three types of robustness tests are shown in Table 4, Columns (1), (2), and (3), respectively. The results of the three types of robustness tests show that the sign and significance of LNPERFDI’s estimated coefficients are consistent with the results in Column (4) of Table 2.

Finally, this study may also face endogenous problems caused by reciprocal causation. Due to the existence of strategic behaviors, the degree of local government competition in a coastal city may be affected by the coastal water pollution of its neighboring coastal cities. For this reason, this paper exploited one-year, two-year, and three-year lags of local competition as the instrumental variables of local government competition in the current year, and a two-stage least squares regression is performed. The reason why we use the lag of local government competition as an instrumental variable is that the lag of local government competition is related to the current local government competition, and the current offshore pollution will not affect the past local competition level, so it can solve the endogenous problems caused by reverse causality. Before the endogeneity test, we conducted the overidentification and weak instrumentals tests to ensure the rationality of instrumental variables. The test results in Column (4) of Table 4 show that the *p*-value of the Hansen J statistic is greater than 0.1 and F statistic is greater than 10. Therefore, the instrumental variables used in this paper are all exogenous, and there is no problem of weak instrumental variables. Then, the Durbin-Wu-Hausman test is used to carry out the endogeneity test. The test results in Column (4) of Table 4 show that the null hypothesis that local government competition is exogenous cannot be rejected. Even so, we still performed a two-stage least squares regression, and the estimated results obtained show that the estimated coefficient of LNPERFDI is still significantly positive.

## 5. Heterogeneity Analysis

### 5.1. Heterogeneity Test Based on Financial Pressures

This research further explores the impact of local government competition on China’s coastal water pollution under different levels of financial pressure. With reference to Bai et al. [13], fiscal pressure is measured by the ratio of the difference between the general public budget expenditure minus the general public budget revenue and GDP. We defined the sample with fiscal pressure above the average fiscal pressure as the high fiscal pressure group. Conversely, the sample below the average fiscal pressure is defined as the low fiscal pressure group. The results of group estimation are shown in Table 5.

Columns (1) and (2) of Table 5 present the estimated results of the impact of local government competition on coastal water pollution in the low and high-pressure groups, respectively. The result shows that the estimated coefficients of LNPERFDI are positive in Columns (1) and (2), but only the estimation results in Column (2) pass the statistical significance test. These outcomes indicate that the impact of local government competition on coastal water pollution is affected by financial pressure. In coastal cities facing high fiscal pressures, the existing official promotion assessment system will encourage local officials to invest financial funds in industries with a high GDP to obtain high economic returns. By contrast, protection and governance of the coastal water environment with low economic returns are easily neglected. In cities facing low fiscal pressures, funds are relatively abundant. In addition to promoting economic growth, there can also be excess funds for coastal water environmental protection and governance. Therefore, Hypothesis 2 is verified.

### 5.2. Heterogeneity Test Based on Period

In different periods, the external environment faced by local development is relatively varied. During the time period examined in this study, two major events occurred in the international and domestic environments facing China. The first is the financial crisis of 2008, and the second is China’s new government assuming power in 2013. These two events would have had an impact on the economic development of local governments and may have affected the estimation results. Therefore, we divided the sample into three stages according to the time points of the two events (i.e., 2004–2007, 2008–2012, and 2013–2017) and performed group regression. The regression results are shown in Table 6.

According to Table 5, the estimated coefficient of LNPERFDI is positive in Columns (1), (2), and (3), but it failed the statistical significance test in Column (1), and pass the statistical significance test at the 10% significance level in Columns (2) and (3). This result shows that the increase in competition among local governments will increase China’s coastal water pollution in 2008–2012 and 2013–2017. The financial crisis in 2008 worsened the external economic environment facing China, thereby increasing the pressure on local governments’ economic development. During the competition, local governments will concentrate additional resources on economic construction to prevent the decline of economic growth. Accordingly, investment in the governance of coastal water pollution will be reduced. In the post-financial crisis era, trade protectionism prevailed, and the external economic environment facing China remained grim. Since the new Chinese government assumed power in 2013, the intensity of environmental governance has continued to increase, but the evaluation mechanism for local officials has not undergone major changes. Moreover, GDP remains the main basis for local officials to be promoted and assessed. There are still situations where coastal local governments sacrifice the coastal water environment to promote economic growth.

## 6. Discussion

Marine water resources are especially important for sustainable agricultural development and food security [42]. In China, the coastal waters environment has become a new area of focus. After the new Chinese government assumed power, the development of the marine economy has gradually become one of the government’s areas of concern. However, the deteriorating coastal water environment has become an obstacle to the sustainable development of the marine economy. This study used a panel data model to explore the impact of coastal local government competition on China’s coastal water environment. The current study contributes to the existing literature in two important ways. First, the existing research on the impact of coastal local governments on coastal pollution has mainly adopted qualitative research methods [26,43,44]. Although Dou and Zhang discussed that GDP-oriented economic growth makes local government competition only focus on economic growth, thereby leading to failure to control coastal water pollution, this conclusion is based on qualitative analysis [45]. The current study uses a panel data model to test the impact of coastal local government competition on coastal pollution. The results show that coastal local government competition will generally increase coastal water pollution. This means that when discussing the control of coastal water pollution, we should focus on what local governments should do and why local governments have not taken effective measures to protect the coastal water environment. That is, we need to focus on the driving mechanism behind local government actions. Second, although fiscal pressure is a key factor influencing local government behavior [46,47], the existing literature on the impact of local government competition on environmental pollution does not consider the importance of fiscal pressure. The present study shows that under different fiscal pressures, the impact of coastal local governments on coastal pollution is heterogeneous, the positive impact of coastal local government competition on coastal water pollution only exists in the high fiscal pressure group. The underlying mechanism may be that coastal cities facing high fiscal pressures have relatively scarce fiscal funds, thereby making local officials allocate fiscal funds to economic construction with the greatest gains in political performance. Thus, it would be crowding out the expenditure on coastal environmental protection and governance.

This research has several limitations that should be addressed in future studies. On the one hand, due to the availability of data, this study used inorganic nitrogen concentration to measure coastal water pollution. However, coastal water pollution is a comprehensive indicator covering multiple pollutants, and inorganic nitrogen concentration alone cannot fully measure coastal water pollution. For the robustness test, although we use chemical oxygen demand to measure coastal water pollution, we still cannot fully measure coastal water pollution. On the other hand, for the robustness test, we attempted to solve the problem of missing variables. We added the proportion of worse than grade V water quality to the control variables, but this indicator did not separate the pollution generated by coastal cities. In the future, if the water pollution indicator in the upper and middle reaches of the basin can be obtained, then additional accurate estimation results can be obtained.

## 7. Conclusions and Policy Implications

China’s political centralization and economic decentralization systems have shaped the economic competition pattern of local governments. Under this system, coastal local governments have invested substantial resources in economic development to gain an advantage in the competition. Moreover, the coastal environment continues to deteriorate. On the basis of theoretical analysis, this study uses panel data from 48 coastal cities in China from 2004 to 2017 to empirically explore the impact of coastal local government competition on coastal water pollution. Our study showed that coastal local government competition increases the concentration of inorganic nitrogen in China’s coastal waters. This conclusion remains valid after conducting various robustness tests. In addition, the impact of coastal local government competition on the coastal water environment will be affected by fiscal pressure. In the high fiscal pressure sample, local government competition will significantly increase the concentration of coastal inorganic nitrogen. However, in the low fiscal pressure sample, the impact of local government competition on the inorganic nitrogen concentration failed the statistical significance test. Furthermore, the impact of coastal local government competition on coastal pollution varies in different periods. Before 2008, the impact of local governments on coastal pollution was positive, but it failed the statistical significance test. From 2008 to 2012 and from 2013 to 2017, coastal local government competition increased coastal water pollution.

The policy implications of our results are as follows. First, in the design of the incentive mechanism for coastal local officials, the coastal water environment should be considered, and coastal waters environmental governance must be incorporated into the official promotion assessment system. The aim of the change assessment system is to promote coastal local governments to compete for growth and improvement of the coastal water environment instead of only for economic growth. Second, the central government should establish a special fund account for coastal water environmental governance through transfer payments, and stipulate that the funds could only be used for coastal water environmental governance. Moreover, coastal cities facing varying financial pressures should be differentiated from one another. The key cities for transfer payments are cities with higher financial pressure to prevent these cities from over-investing financial funds into economic construction in local government competition.

## Figures and Tables

**Figure 1 ijerph-17-06862-f001:**
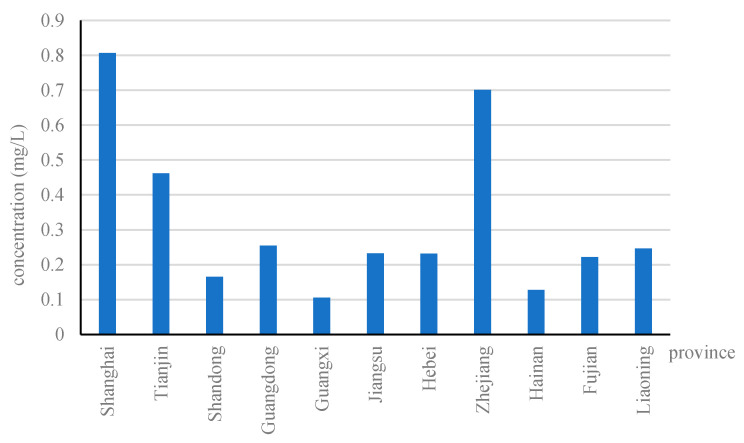
Average concentration of inorganic nitrogen in 11 coastal provinces.

**Table 1 ijerph-17-06862-t001:** Summary statistics.

Variables	Variable Definition	Obs	Mean	SD	Min	Max
LNORGANICN	Inorganic nitrogen concentration (mg/L)	655	0.283	0.301	0.003	2.271
LNPERFDI	Logarithm of per capita actual utilization of FDI (yuan/person)	655	6.559	1.432	1.229	9.774
LNPERGDP	Logarithm of GDP per capita (Thousands of yuan)	655	10.396	0.639	8.500	12.784
LNPERGDP2	the quadratic term of the logarithm of GDP per capita	655	108.486	13.260	72.250	163.418
LNDENSITY	Logarithm of population density (person/km^2^)	655	7.637	0.712	5.557	9.908
LNPATGRANTED	Logarithm of total number of patent grants (piece)	655	7.215	1.820	1.609	11.449
URBANIZATION	Urbanization (%)	655	48.155	20.782	10.142	100.000
SECONDGDP	Proportion of secondary industry to GDP (%)	655	48.881	9.470	18.140	82.280
LNMGDP	Logarithm of gross ocean product (100 million yuan)	655	7.959	1.240	2.226	9.809

**Table 2 ijerph-17-06862-t002:** Estimation results of the entire sample.

Variables	(1)	(2)	(3)	(4)
	LNORGANICN	LNORGANICN	LNORGANICN	LNORGANICN
LNPERFDI	0.061 ***	0.020 **	0.050 ***	0.023 **
	(0.009)	(0.010)	(0.009)	(0.010)
LNPERGDP		−0.774 ***		−1.121 ***
		(0.276)		(0.343)
LNPERGDP2		0.039 ***		0.052 ***
		(0.013)		(0.016)
LNDENSITY		0.061 ***		0.080 ***
		(0.015)		(0.015)
LNPATGRANTED		0.060 ***		0.004
		(0.008)		(0.008)
URBANIZATION		−0.001		0.003 ***
		(0.001)		(0.001)
SECONDGDP		0.000		0.003 **
		(0.001)		(0.002)
LNMGDP		−0.026 **		0.008
		(0.012)		(0.033)
Constant	−0.119 **	3.267 **	−0.304 ***	4.872 ***
	(0.049)	(1.397)	(0.062)	(1.762)
Province effect	No	No	Yes	Yes
Year effect	No	No	Yes	Yes
N	655	653	655	653
Adj R2	0.084	0.204	0.432	0.483

Note: Figures in () are robust standard error; ***, and ** indicate significance at the 1%, and 5% levels, respectively.

**Table 3 ijerph-17-06862-t003:** Estimation results of replacing proxy variables.

Variables	(1)	(2)	(3)	(4)
	COD	LNORGANICN	LNORGANICN	LNORGANICN
LNPERFDI	0.108 ***			
	(0.038)			
PROFDI		0.013 **		
		(0.006)		
DUSAGE			0.031 **	
			(0.015)	
DUMAGE				0.029 *
				(0.015)
LNPERGDP	−2.155 **	−1.061 ***	−1.066 ***	−1.097 ***
	(0.849)	(0.310)	(0.328)	(0.317)
LNPERGDP2	0.092 **	0.052 ***	0.051 ***	0.053 ***
	(0.038)	(0.015)	(0.016)	(0.015)
LNDENSITY	0.203 ***	0.083 ***	0.082 ***	0.078 ***
	(0.048)	(0.015)	(0.016)	(0.016)
LNPATGRANTED	−0.065 ***	0.014	0.009	0.010
	(0.024)	(0.009)	(0.008)	(0.008)
URBANIZATION	0.007 ***	0.004 ***	0.004 ***	0.004 ***
	(0.002)	(0.001)	(0.001)	(0.001)
SECONDGDP	0.019 ***	0.002	0.003 **	0.003 *
	(0.005)	(0.002)	(0.001)	(0.002)
LNMGDP	−0.019	0.020	0.010	0.012
	(0.061)	(0.033)	(0.033)	(0.033)
Constant	9.548 **	4.275 ***	4.509 ***	4.689 ***
	(4.368)	(1.577)	(1.679)	(1.629)
Province effect	Yes	Yes	Yes	Yes
Year effect	Yes	Yes	Yes	Yes
N	557	653	653	653
Adj R2	0.247	0.481	0.481	0.480

Note: Figures in () are robust standard error; ***, **, and * indicate significance at the 1%, 5%, and 10% levels, respectively.

**Table 4 ijerph-17-06862-t004:** Results of other robustness tests.

Variables	(1)	(2)	(3)	(4)
	LNORGANICN	LNORGANICN	LNORGANICN	LNORGANICN
LNPERFDI	0.023 **	0.019 **	0.027 **	0.037 **
	(0.010)	(0.009)	(0.011)	(0.015)
LNPERGDP	−1.136 ***	−1.072 ***	−1.164 ***	−0.945 *
	(0.349)	(0.329)	(0.339)	(0.498)
LNPERGDP2	0.053 ***	0.051 ***	0.054 ***	0.043 *
	(0.017)	(0.016)	(0.016)	(0.024)
LNDENSITY	0.078 ***	0.075 ***	0.083 ***	0.092 ***
	(0.015)	(0.014)	(0.016)	(0.018)
LNPATGRANTED	0.005	0.005	0.001	−0.003
	(0.008)	(0.008)	(0.008)	(0.010)
URBANIZATION	0.003 ***	0.003 ***	0.003 ***	0.003 ***
	(0.001)	(0.001)	(0.001)	(0.001)
SECONDGDP	0.003 **	0.003 *	0.003 **	0.004 **
	(0.002)	(0.001)	(0.002)	(0.002)
LNMGDP	−0.026	0.010	0.005	−0.009
	(0.051)	(0.032)	(0.033)	(0.076)
BASINPOLL			0.004 ***	
			(0.001)	
Constant	5.157 ***	4.630 ***	5.012 ***	4.681 *
	(1.785)	(1.690)	(1.735)	(2.537)
Province effect	Yes	Yes	Yes	Yes
Year effect	Yes	Yes	Yes	Yes
N	625	653	653	517
Adj R2	0.443	0.500	0.487	0.498
DWH *p*-value				0.382
Hansen J *p*-value				0.336
F statistics				179.550

Note: Figures in () are robust standard error; ***, **, and * indicate significance at the 1%, 5%, and 10% levels, respectively. DWH is the Durbin-Wu-Hausman test. F statistics are excluded-instruments F statistics.

**Table 5 ijerph-17-06862-t005:** Estimated results for the low and high fiscal pressure groups.

Variables	(1)	(2)
	Low	High
LNPERFDI	0.020	0.039 **
	(0.014)	(0.018)
LNPERGDP	−1.256 **	−1.164
	(0.508)	(0.923)
LNPERGDP2	0.060 **	0.050
	(0.025)	(0.044)
LNDENSITY	0.082 ***	0.085 ***
	(0.019)	(0.030)
LNPATGRANTED	0.005	−0.016
	(0.011)	(0.015)
URBANIZATION	0.003 ***	0.004 ***
	(0.001)	(0.001)
SECONDGDP	0.003 *	0.006 *
	(0.002)	(0.003)
LNMGDP	0.017	−0.160
	(0.033)	(0.145)
Constant	5.432 **	6.468
	(2.596)	(4.831)
Province effect	Yes	Yes
Year effect	Yes	Yes
N	367	286
Adj R2	0.457	0.537

Note: Figures in () are robust standard error; ***, **, and * indicate significance at the 1%, 5%, and 10% levels, respectively.

**Table 6 ijerph-17-06862-t006:** Estimated results of the periods before 2007, 2007–2012, and 2013–2017.

Variables	(1)	(2)	(3)
	2004–2007	2008–2012	2013–2017
LNPERFDI	0.003	0.042 *	0.033 *
	(0.022)	(0.022)	(0.018)
LNPERGDP	−1.156	−0.990	−1.823 **
	(0.726)	(0.705)	(0.837)
LNPERGDP2	0.058	0.045	0.082 **
	(0.036)	(0.034)	(0.038)
LNDENSITY	0.046 **	0.108 ***	0.114 ***
	(0.022)	(0.026)	(0.038)
LNPATGRANTED	0.004	−0.003	−0.015
	(0.015)	(0.013)	(0.021)
URBANIZATION	0.002 **	0.003 ***	0.005 ***
	(0.001)	(0.001)	(0.002)
SECONDGDP	0.002	0.004	0.006
	(0.003)	(0.003)	(0.004)
LNMGDP	0.017	0.206	−0.203
	(0.039)	(0.179)	(0.168)
Constant	5.052	2.532	10.734 **
	(3.578)	(3.654)	(4.834)
Province effect	Yes	Yes	Yes
Year effect	Yes	Yes	Yes
N	183	235	235
Adj R2	0.418	0.469	0.528

Note: Figures in () are robust standard error; ***, **, and * indicate significance at the 1%, 5%, and 10% levels, respectively.

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
