# Peer review of "Does Coastal Local Government Competition Increase Coastal Water Pollution? Evidence from China"

_ijerph, 2020, doi:10.3390/ijerph17186862_

Round 1

Reviewer 1 Report

The paper “Does Costal Local Government Competition Increase Costal water Pollution? Evidence from China” has shown a new approach to analyses of serious and complex problem of costal water contamination. As Authors presented local government competitions to promote economic growth play the important role in the deteriorating of coastal water environment. In the paper heterogeneity analysis have been performed and described, which using different factors such as: financial pressures, urbanization, promotion incentives of costal city officials or different periods and inorganic nitrogen and chemical oxygen demand as the factors of coastal water pollution. The Authors put two hypotheses: 1. “Coastal local government competition will increase coastal water pollution”: and 2. “The impact of coastal local government competition on coastal water pollution will vary in cities with high fiscal pressure and cities with low fiscal pressure”. Both of hypotheses were verified and supported. The paper is very interesting and generally good prepared, but before publication I suggest some additions and improvements:

Technical corrections

- page 2 line 80-82: Difference in organization of the paper:

“Section 4 provides the discussion and Implications” but in text is “Empirical Results” Is this the same?

“Section 5 presents the conclusion” but in text is “Heterogeneity Analysis”

In this fragment of the paper is not information about section 6 and 7.

- page 13 line 462: “6. Conclusion and Policy Implications” and page 14 line 490: “7. Discussion” Section Discussion should be before conclusions

- page 2 line 44: twice “be”

- page 2 line 54: “GDP” – is not clearly explained what the abbreviation means.

- page 3  line 107 Section “2.1.2. Research…..”  - space after the text above

- page 3 line 110 “….have emerged. coastal water …” should be “….have emerged. Coastal water …

- page 4 line 156: cited literature [31]  before literature [29]  (page 4 line 165)

- page 5 formula 1 – is not explained what β (beta) means

- page 6 Figure 1 – there is not concentration unit

- page 13 line 448: “….in ColumnS (1), (2), ….” should be “….in Columns (1), (2), …

- page 1 line 37; page 3 line 97, 112;   page 4 line 146, 159, 165; page 6 line 251, 260; page 10 line 371; page 14 line 498, 507: space before cited literature […]

- page 15 line 566 – names of authors

- page 15 line 570  - name of author – space?

Reviewer 2 Report

This is an interesting applied paper  focused on the spillover effects of economic competition among local governments on the coastal water pollution. In this study

prefecture-level data have been used to analyze coastal water pollution, since it reflect the importance of local competition in China's coastal water pollution control. Concerning methodology, authors used a fixed-effects panel regression model.

The topic is very interesting and may have relevant policy implications at local level. I consider the paper connected with the overall philosophy of the Journal.

My recommendation is to ACCEPT the paper subject to MINOR CHANGES for the reasons described in continuation:

  • In paraghaph 2.1.2 “Research on the causes and governance of coastal water environment”, It would be possible to quantify the coastal water pollution in China?
  • Authors should describe more in detail the source of data used.
  • Some references omitted must be included, for example I suggest authors to include the following papers:
    • Benedetti, I., Branca, G., & Zucaro, R. (2019). Evaluating input use efficiency in agriculture through a stochastic frontier production: An application on a case study in Apulia (Italy). Journal of Cleaner Production, 236, 117609.
  • Among the Robustness Checks, I suggest authors to detect the presence of endogeneity in the model and carry out a specific test.
  • A careful round of editing is needed since several typos are present in the manuscript. For example: in line 110 “coastal water” need to be in capital letter.
